# On the Role of Informal vs. Formal Context of Language Experience in Italian–German Primary School Children

Mariapaola Piccione [1], Maria Francesca Ferin [2], Noemi Furlani [3], Miriam Geiß [2], Theodoros Marinis [2] and Tanja Kupisch [2,*]

1   Institute of Linguistics/Romance Languages, University of Stuttgart, 70174 Stuttgart, Germany; mariapaola.piccione@ling.uni-stuttgart.de
2   Department of Linguistics, University of Konstanz, 78464 Konstanz, Germany; maria.ferin@uni-konstanz.de (M.F.F.); miriam.geiss@uni-konstanz.de (M.G.); t.marinis@uni-konstanz.de (T.M.)
3   Department of Linguistics, University of Cologne, 50923 Köln, Germany; nfurlani@uni-koeln.de
*   Correspondence: tanja.kupisch@uni-konstanz.de

**Abstract:** This study focuses on the contexts of language experience in relation to language dominance in eighty-seven Italian–German primary school children in Germany using the MAIN narrative task. We compare current language experience in the heritage language (Italian) and the majority language (German) in both formal and informal settings, and we examine the respective impact on micro- and macrostructure measures, including different language domains. Some previous findings emphasized the importance of language experience in formal contexts. By contrast, our results suggest that, in particular, language experience in informal contexts determines vocabulary and fluency in the heritage and majority language, while there are no effects of exposure on syntactic complexity. Furthermore, while the younger children are relatively balanced, the older children are more dominant in the societal language. Our findings imply that the use of the minority language in informal contexts should be encouraged to promote its development and maintenance.

**Keywords:** bilingualism; language dominance; language experience; narratives; proficiency





## 1. Introduction

Language dominance is an intuitively clear concept, but it is scientifically hard to capture. It usually refers to differences in the amount of use and proficiency between two languages of an individual speaker. With respect to bilingual children, language dominance has been operationalized either based on (i) reported language experience using parental questionnaires or (ii) relative proficiency measures using language tasks (Treffers-Daller and Korybski 2015). Parental questionnaires are an *indirect measure* of language experience and usually include questions on language exposure and use within and outside the home. They typically result in classifications of bilinguals as more or less balanced (e.g., Birdsong et al. 2012; Anderson et al. 2021). While questionnaires are a quick measure to assess dominance, they do not determine how children perform with respect to various linguistic abilities (e.g., grammar, vocabulary, and fluency). The latter must be assessed on the basis of language tasks, i.e., with a *direct measure* of language dominance. For practical reasons, such as time constraints and availability of assessment material, linguistic abilities are often measured based on a single language domain, e.g., lexicon or morpho-syntax. However, differences in language skills in bilingual children's two languages are not always visible across the board (e.g., Paradis and Genesee 1996), which underlines the need to assess children's abilities across language domains.

In the present study, we combine an *indirect*, questionnaire-based measure that taps into language experience with a *direct*, speech-data-based measure of dominance (Treffers-Daller 2019). We use a narrative task to directly measure a range of language skills.

Data come from Italian–German bilingual primary school children in Germany, who have acquired Italian as their heritage language (HL) and German as the majority language (ML) of the society. We focused on this group because a considerable number of people living in Germany are of Italian origin (930,000 people according to the Statistische Bundesamt Destatis (2021)). Moreover, Italian as a heritage language is supported by the active work of associations, consulates, Italian Cultural Institutes, and embassies, which promote its development and maintenance among Italian speakers and families in Germany. For both language experience and proficiency measures, we calculated "dominance differentials" to investigate how the HL and ML interact with each other. We are particularly interested in whether language experience in formal and informal settings interacts differently with relative language proficiency, and which language domains are more easily affected by changes in language experience.

## 2. Background

### 2.1. Dominance and Schooling in the ML

Language dominance has been the object of a large body of literature investigating child and adult *heritage speakers*. At the outset of language acquisition in bilingual speakers, there are three possible language dominance scenarios. Children may be relatively balanced (e.g., Müller and Hulk 2001; Kupisch 2006, 2007) or they may start out as dominant in the ML (e.g., Lanza 2004; Kehoe et al. 2004; Kupisch 2006, 2007). These two cases are typical for *one parent one language* contexts. Alternatively, they may be dominant in their HL (e.g., Papastefanou et al. 2019, 2021) because they stay at home where the only language used is the HL, and they are exposed to the ML only later when attending daycare or school; countries may differ as to when children typically do so.

Language contact with the HL is typically restricted to the home, and when children start school, their exposure to the national language, the major medium of instruction, increases. The beginning of school is therefore a crucial point in determining the trajectories of language experience in the HL and ML: the language used in school will benefit from wider input, including an academic register that is not counterbalanced in the HL (Montanari et al. 2018). School-aged children in particular can benefit from the persistence of stable patterns of the HL use over the years (*consistency*) and the opportunity to receive formal education in the HL (*continuity*) in bridging the gap between HL and ML input (Caloi and Torregrossa 2021).

Many studies have shown that entering primary school can cause changes in dominance for those children who are not yet dominant in the societal language, but balanced or dominant in their HL. In a study by Rinker et al. (2022), Italian–German kindergarten children had balanced vocabulary skills, whereas elementary school children had a larger vocabulary in German than in Italian. Kupisch et al. (2021) investigated the perceived accents of Russian–German bilingual children in Germany, comparing preschools (ages 4–6) and school children (ages 7–9). The younger children tended to have stronger accents in German (their ML) than in Russian (their HL), while the older children were more accented in Russian. Papastefanou et al. (2019, 2021) used parental questionnaires together with a range of language and reading measures. The parental questionnaires showed that most children younger than four years old were dominant in their HL, but during primary school, they were dominant in their ML, indicating that the turning point for dominance was school entry. Direct measures of language and reading (expressive vocabulary, phonological awareness, morphological awareness, morphosyntactic abilities, reading accuracy of words and non-words, and reading comprehension) confirmed a dominance in the ML during primary school. This remained stable within the first four years of primary school except in the production of derivational morphology, where the shift in dominance from the HL happened a little later (between Grades 1 and 3).

In summary, most studies assessing language dominance have focused on preschool children, where a variety of dominance scenarios can be found, depending on language use in the home. A few studies have focused on language dominance during primary school,

highlighting shifts towards the societal majority language. It remains unclear whether the shift happens as a result of formal and/or informal language experience and whether it affects all language domains equally.

## 2.2. Reported Language Experience

*Language experience* has often been operationalized as "language use", "exposure", or "input" and measured using parental questionnaires (Carroll 2017). In this study, we focus on the opposition between *formal* (e.g., at school and through reading) and *informal* (e.g., at home and on the playground) using *experience* as an overarching term.

Language experience has been measured in different ways, and questionnaires have become more refined over time. Gathercole and Thomas (2005) have classified children based on their parents' native language and the medium of education in school (Welsh-only vs. Welsh–English). Unsworth (2013, 2015) has distinguished the *current* amount of exposure and the *cumulative* amount of exposure, both calculated as relative exposure to the HL and ML. Additional measures were (1) *richness* (language use in extra-curricular activities), (2) *native* (nativeness/non-nativeness of the interaction partners), (3) number of exclusive speakers of a certain language at home, and (4) number of different HL and ML speakers at home. Mattheoudakis et al. (2016) have compared informal input at home vs. formal input at school, taking both current and cumulative exposure into account.

It has been consistently shown that the amount of language experience is crucial for predicting acquisition outcomes, especially in the heritage language (Gathercole and Thomas 2005; Armon-Lotem et al. 2011; Chondrogianni and Marinis 2011; Paradis 2011; Unsworth 2013, 2015; Paradis and Jia 2017), although a smaller number of studies has shown that in some cases, even a weaker language can boost the development of the stronger language (Kupisch 2006). While parents' use of the HL is highly beneficial for the development of the HL (Sorenson Duncan and Paradis 2020; Gathercole and Thomas 2005), it does not seem to affect the development of the ML, either positively or negatively.

There is some indication that formal exposure has a special role. Formal exposure may drive a shift from the HL to the ML if the latter is the language of instruction in school (e.g., Jia and Paradis 2015). In the ML, language experience in formal settings correlates with vocabulary size, narrative skills, morphological accuracy, and syntactic complexity (e.g., Golberg et al. 2008; Paradis 2011). Similarly, in the HL, formal exposure has beneficial effects on vocabulary and syntactic complexity (e.g., Bayram et al. 2019; Kupisch and Rothman 2018; Caloi and Torregrossa 2021; Mattheoudakis et al. 2016). At the same time, there is evidence that informal interactions can also boost HL vocabulary and morphosyntax (e.g., Schmid and Karayayla 2020, in the case of adults).

In summary, there is a lot of evidence that formal language experience is central to language development in both ML and HL, while it remains somewhat unclear whether, for some domains of language to develop, informal language experience might be sufficient. Moreover, it remains unclear how exposure to one language affects the other language. Our study compares formal and informal contexts of language experience, in the HL vs. ML and their effects across linguistic domains.

## 2.3. Narrative Abilities

Narrative abilities are essential to social interactions (McCabe 1996), the development of literacy (e.g., Pearson 2002), and scholarly performance in general. Bilingual children show narrative abilities in both languages from preschool age (e.g., Gutiérrez-Clellen 2002; Pearson 2002; Uccelli and Páez 2007).

Narrative abilities have been operationalized in terms of microstructural and macrostructural ones. Microstructure is language-specific (Gagarina et al. 2019b) and covers various proficiency measures, e.g., morpho-syntax, vocabulary, phonological development, and fluency (e.g., hesitations or speech rate), language choice and code-mixing, as well as more global parameters, such as translation or sentence repetition, or the sheer quantity of speech. Lexical measures (word types, type-token ratio, and lexical diversity/breadth) are very common

(Treffers-Daller 2011), but the use of lexical measures alone may reveal only part of the picture because bilingual children are known to differ from monolinguals, especially in their vocabulary development, though perhaps less in their morpho-syntactic development. While using a variety of measures is desirable, it may be hindered by practical constraints, such as time.

Several recent studies analyzing microstructure in bilingual children have been based on the MAIN task (Gagarina et al. 2012, 2019a; https://main.leibniz-zas.de (accessed on 9 January 2020)), which allows the extraction of multiple measures without recurring to long and tiresome assessments. For example, in a study of preschool Russian–Norwegian bilinguals in Norway, Rodina (2017) demonstrated an effect of exposure for various microstructure measurements, including communication units (main and subordinate clauses), number of word tokens, MLU, and total number of verbs and nouns. The bilingual children scored higher in the ML than in the HL. Also based on the MAIN, Altman et al. (2016) investigated language dominance in bilingual preschool children with L1 English and L2 Hebrew assessing the total number of word tokens, number of different words, mean length of C-units, longest C-units, and morphosyntactic errors. The children scored higher in their L1 than in their L2.

The second component of narrative ability is macrostructure, i.e., the organization of the content of a narrative, also referred to as "story grammar". The story grammar includes a setting, an initiating event, internal responses and plans, attempts, consequences, and reactions (Stein and Glenn 1979). The combination of these elements constitutes a full episode with *Goal*, *Attempt*, and *Outcome* (e.g., McCabe and Peterson 1984), which is shared across languages. Studies on macrostructure in bilingual children have focused primarily on whether children's macrostructure improves with age and whether it improves differently across languages. Below, we summarize studies based on the MAIN task, which measures macrostructure by means of (A) story grammar elements, (B) story structural complexity, (C) internal state terms, and (D) story comprehension. The four components (A–D) can be examined separately, and some studies only used a subset.

Studies on macrostructure in bilingual children have shown partially contradictory results. Bohnacker et al. (2021) found that macrostructure improved with age in both languages of four-to-seven-year-old Turkish–Swedish bilingual children. Similar findings were reported in a study by Tribushinina et al. (2022) of five-to-eleven-year-old Indonesian–Dutch bilinguals. There was a positive cross-language carry-over effect in story grammar and structural complexity between the languages, indicating that narrative skills can be transferred between languages. Gagarina (2016) found that Russian–German bilingual children in Germany improved their macrostructure skills between ages 3 and 7, but no more between 7 and 9 years, except in story complexity in Russian. It is possible that age effects in the German language were absent because the children had already reached high skills before age 7. In contrast to Gagarina (2016), Bohnacker (2016) found an effect of age but not of language in Swedish-English bilingual children between 5 and 7 years old. Moreover, Kunnari et al. (2016) found neither an effect of age nor of language in five-to-six-year-old Finnish–Swedish simultaneous bilingual children. Differences across studies could depend on extra-linguistic factors, such as the social status of the respective minority language. Furthermore, some aspects of macrostructure development seem to depend on language experience and microstructure skills. For example, Lindgren and Bohnacker (2022) showed that macrostructural skills developed between 4 and 6 years in the two languages of German–Swedish bilinguals, but story structure scores were lower in their HL, where the children also had smaller vocabularies.

In summary, macrostructure skills increase with age and there may be cross-language carry-over effects. Such cross-over effects are expected because, unlike microstructure elements, narrative abilities involve cognitive processes that do not differ across languages. However, the effect of age is not consistent across studies and languages and may not occur in both languages. Although language exposure can matter, it seems to be less crucial for macrostructure development than for microstructure development.

## 3. Research Questions and Predictions

As mentioned above, previous research has pointed out that dominance is unstable, leading to shifts usually from the HL to the ML. The turning point during which this shift happens is somewhere between kindergarten and the end of primary school, where the ML is typically the medium of instruction. However, relatively little is known about how exactly dominance evolves during the school years and how the contexts (formal vs. informal) of language experience contribute to changing dominance in the various domains of language. We investigate these points based on data from Italian–German bilingual children during primary school. Our research questions are as follows:

RQ1: What are bilingual children's formal and informal experiences during primary school?

RQ2: How do the contexts of language experience (formal and informal) correlate with proficiency in different linguistic domains (microstructure: vocabulary, fluency, morphosyntax, and syntactic complexity) and macrostructure?

RQ3: Is there a change in language experience and microstructure and macrostructure skills, pointing to a change in dominance across age groups? If so, how is it reflected across linguistic domains?

First, bilingual children in Germany usually attend German schools and are exposed to German outside of their homes and sometimes within their homes as well, while exposure to Italian is limited to the home in most cases. Therefore, we expect that primary school children will have more language experience in German, especially in formal contexts, but we expect little variability within the two contexts across age groups (RQ1).

Second, previous findings have shown positive correlations between language experience and various language skills in the HL and ML (e.g., Armon-Lotem et al. 2011; Chang 2016; Gathercole and Thomas 2005; Lloyd-Smith et al. 2020). The few studies that have compared formal and informal language use have found conflicting results, with some showing that formal language experience in the HL is a strong predictor for HL grammar (Bayram et al. 2019) and vocabulary skills in both HL and ML (Mattheoudakis et al. 2016); others showed effects of informal experience only for HL vocabulary and morphosyntax in adults (Schmid and Karayayla 2020). We expect to find an effect of formal and potentially informal language experience on HL and ML microstructure measures, particularly for vocabulary and syntax (RQ2). Since macrostructure is more dependent on general cognitive skills, we expect that it will be less affected by experience (RQ2).

Third, schooling in German and the development of reading and writing skills should lead to an increase in formal exposure to German over time (RQ1, RQ3). The new environment can further cause changes in attitudes towards the HL and ML and a language shift in the home, thus also affecting informal use of the HL and ML. We therefore expect an increase in both formal and informal language experience of German as an ML and a decrease in Italian as an HL. Similarly, we expect proficiency to grow faster in the ML than in the HL, resulting in a widening gap in microstructure, which is not necessarily mirrored in macrostructure skills.

## 4. Methods

### 4.1. Participants

Eighty-seven Italian–German bilingual kids (Biki) living in Germany participated in this study. The children's ages ranged from 6 to 9 years, and they were divided into four groups accordingly (*Biki6*[1], *Biki7*, *Biki8*, and *Biki9*). Participants were recruited through flyers, emails, social media, Italian associations in Germany, consulates, and schools. The children either acquired both languages from birth (2L1 *n* = 50) or Italian from birth and German as an early second language (eL2 *n* = 37), with an Age of Onset (AoO) ranging between 1 and 7 years (mean: 2;9). Children who acquired languages other than Italian or German before age 7 were not included. Table 1 provides an overview of the participants.

**Table 1.** Overview of participants.

| | Total | N Female (%) | Mean Age in Years (SD) | N 2L1 | N eL2 (AoO Ger: Mean, SD) |
|---|---|---|---|---|---|
| Biki6 | 24 | 15 (63%) | 6.5 (0.3) | 15 | 9 (2.75, 1.17) |
| Biki7 | 22 | 11 (50%) | 7.5 (0.3) | 15 | 7 (3.04, 0.80) |
| Biki8 | 18 | 12 (67%) | 8.7 (0.2) | 6 | 12 (2.97, 1.46) |
| Biki9 | 23 | 7 (30%) | 9.5 (0.3) | 14 | 9 (3, 1.71) |

Most children (*n* = 46) had two Italian-speaking parents who were first-generation immigrants; in a few families (*n* = 9), one or two parents were second-generation immigrants and spoke Italian as an HL. In 32 families, one parent spoke German, while the other one spoke Italian either in the first (*n* = 26) or second generation (*n* = 6). Regarding socioeconomic status (SES), the majority of the children's parents had a university degree (mothers: 71% and fathers: 77%) or a high-school degree (German *Realschule* or *Abitur*, Italian *diploma professionale* or *maturità*) (mothers: 24% and fathers: 20%). Only eight parents did not have a high school diploma.

*4.2. Materials*

Parents were asked to fill in a questionnaire on their children's language experience using the SoSci Survey (Leiner 2019). The questionnaire included six parts: (i) general information about the child and (ii) the family, (iii) language experience in the family, child education without (iv) and with (v) the pandemic, and (vi) language during spare time[2].

Language data were collected using the *Cat* and the *Dog* stories of the MAIN task (Gagarina et al. 2012, 2019a; Levorato and Roch 2020; see https://main.leibniz-zas.de (accessed on 9 January 2020)). The two stories have comparable structures, events, and main characters, and are comparable in length (165 and 167 words). The Italian and German stories were told by an Italian and a German native speaker, respectively.

*4.3. Procedure*

This study was conducted online during the pandemic (2021–2022). Parents first completed the online questionnaire that ensured that the children met the criteria for participation. Parents were then contacted to schedule one session for data collection in Italian and one for data collection in German. Parents were instructed that a laptop or tablet was necessary to ensure high audio quality for the recordings. They had to be present at the beginning of each session to help the children with potential technical difficulties. The experimenter conducted the task via Zoom using the 'Share Screen' option and parents were instructed to use the full-screen mode on their device. During testing, parents were asked not to help or interfere. The tasks were presented as a game; children participated in a training program at a fictional detective school (Figure 1).

Children retold the story in both Italian and German in separate sessions, alternating the *Cat* and *Dog* stories. Those who encountered the *Cat* story in Italian in the first session listened to the *Dog* story in German in the second session, and vice versa. The two sessions were conducted at least one week apart with two different native speakers of the respective languages.

In each session, children were instructed to look at the pictures, listen carefully to the story, and retell it to the experimenter, who pretended that she could not see or hear the story. Following the MAIN procedure, the six pictures appeared one by one on the screen so that children could familiarize themselves with the story. After this, the child listened to the narrating voice, and the pictures appeared pairwise. After the story, the experimenter asked the child to retell the story while looking at the pictures, which were again presented pairwise, and finally, the child was asked 10 comprehension questions (Gagarina et al. 2019a) about the story components of each event (Goal, Attempt, Outcome, and Reaction). The retelling and the comprehension questions were recorded by the experimenter using Audacity (Audacity Team 2021). The task lasted for approximately 10 min.

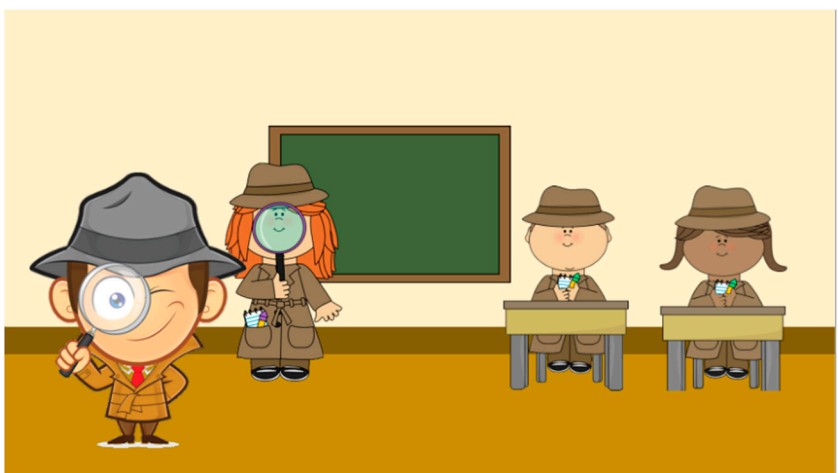

**Figure 1.** Detective school setting presented during the tasks.

*4.4. Data Analysis*

For this study, we used the parental questionnaire to investigate the areas of formal vs. informal context of language experience, including more quantitative (e.g., amount of language use and length of stays in Italy) and more qualitative questions (e.g., people involved in the interactions and types of activities) (Table 2). The scoring was rescaled to a maximum of 10 points for each of the two sections. Each score was calculated for German and Italian separately (see Supplementary Materials Section S2 for details).

**Table 2.** Areas of parental questionnaire investigated for this study.

| FORMAL | INFORMAL |
|---|---|
| - Amount of Italian/German experience in formal settings (e.g., school, kindergarten, and Italian courses) | - Amount of Italian/German experience in informal settings (e.g., with family and friends)<br>- Length of periods in Italy |
| - Number of different people involved in interactions (e.g., teachers and school peers)<br>- Type of formal engagement (books and audiobooks) | - Number of different people involved in interactions (e.g., family members and best friends)<br>- Freetime activities (music, TV, and videogames) |

The narratives were transcribed using CLAN in the CHAT format, following the CHILDES manual (MacWhinney 2000). All transcripts were transcribed by native speakers of German and Italian and cross-checked by a second transcriber, who was also a native speaker; disagreements were discussed and resolved. The transcriptions were analyzed for microstructure and macrostructure.

We employed four microstructure measures to operationalize proficiency in vocabulary, fluency, morpho-syntax, and syntactic complexity. Using the KIDEVAL command in CLAN, we extracted Moving Average Type Token Ratio (MATTR) (a measure of lexical diversity independent of text length) for vocabulary (Zenker and Kyle 2021), speech rate (in words/second) for fluency, calculated by dividing the total number of words by the duration of each narrative, excluding pauses between utterances, and Mean Length of Utterances (MLU) in words as a measure of morpho-syntax. For syntactic complexity, we further calculated a complexity index (CI) by summing independent and dependent clauses together and then dividing them by the number of independent clauses (Schneider et al. 2005). For macrostructure, we analyzed parts B and D of the MAIN scoring sheet (Gagarina et al. 2019a; Levorato and Roch 2020). Part B measures structural complexity. Each story is composed of three episodes; each with three elements: an explicit mention of the character's Goal, an Attempt to fulfill, and the Outcome of the action. Following

the literature, if children mentioned all three-episode elements, they scored 3 points. For only the Goal, the Goal/Attempt combination, or the Goal/Outcome combination, they scored 2 points. If only Attempt and Outcome were described, they scored 1 point. The highest possible score was 9 (3 complete episodes). Part D rated the children's answers to the comprehension questions (maximum of 10 points).[3] We considered dominance as gradient rather than categorical (Birdsong 2015) and, for this reason, for each micro- and macrostructure measurement, we calculated a differential score by subtracting the Italian from the German score: a positive differential score indicates dominance in German, and a negative differential score indicates dominance in Italian. We used the differentials as a continuous variable for our statistical analyses.

To address our RQs, we used linear models (*lm*) included in the R package *stats* (R Core Team 2021). The summary function of the *lmerTest* package (Kuznetsova et al. 2017) was used to obtain *p*-values. For RQ1, we fitted linear models with either 'German formal language experience' (GE-for), 'German informal language experience' (GE-inf), 'Italian formal language experience' (IT-for), 'Italian informal language experience' (IT-inf), 'formal differentials' (formal diff.), or 'informal differentials' (informal diff.). We used 'age in months' as a continuous independent variable. To address RQ2, we calculated separate linear models with one of the proficiency measures of each language as the dependent variable each time. We used 'German/Italian formal language experience' and 'German/Italian informal language experience' as fixed predictors to investigate the effects of German/Italian language experience on the respective proficiency measures. To address RQ3, we fitted linear models using the differentials of each proficiency measure as the dependent variable. We used 'age in months', 'formal differentials', and 'informal differentials' as independent variables. The most complex models we considered consisted of two two-way interactions between age and the other two predictors. Non-significant interactions were removed, but non-significant main effects were kept.

## 5. Results

The results are reported as follows: For each RQ, we first report the descriptive statistics, and then the results of the statistical models are presented one by one, with the *p*-values (see Supplementary Materials Section S4 for the details of the statistical analyses).

### 5.1. Formal and Informal Language Experience

Table 3 and Figure 2 summarize the formal and informal language experience for each group in German and Italian. In formal contexts, all age groups used more German than Italian, although this contrast is less pronounced in Biki8. In informal contexts, Biki6 and Biki8 used more Italian than German, while Biki7 and Biki9 used more German than Italian.

For formal language experience, the statistical analysis showed no effect of age as a continuous variable (GE-for: $p = 0.18$; IT-for, $p = 0.26$) (Supplementary Materials Section S4, Table S4.2). However, since at the descriptive level Biki8 seemed to have more formal language experience in Italian than the other three groups, we ran additional models with age group as a categorical predictor to confirm this trend. The results confirm that Biki8 behaves differently from the other three groups. Indeed, Biki8 had significantly less formal German experience than Biki6 ($p = 0.03$) and had less formal German experience than Biki7 ($p = 0.06$) (approaching significance). Conversely, Biki8 has more formal Italian experience than Biki6 ($p = 0.04$) and Biki7 ($p = 0.04$) (Supplementary Materials Section S4, Table S4.3). To understand the difference in formal language experience between Biki8 and the other age groups, we considered the individual data and identified 7 out of 18 children (39%) in Biki8 who attended schools with a bilingual curriculum, whereas the percentage of children in bilingual schools in the other groups ranged from 9% to 22%[4]. Among the seven children of the Biki8 group, six were reported to speak each language half of the time with their teachers and one was reported to speak more Italian than German. For interactions with their peers at school, four out of the seven children used Italian and German equally, two

used more Italian, and one used more German. In the other age groups, children reported using German and Italian or more German when interacting with teachers or peers.

**Table 3.** Mean scores (SD) for formal and informal language experience (German and Italian).

|  |  | Biki6 (*n* = 24) | Biki7 (*n* = 22) | Biki8 (*n* = 18) | Biki9 (*n* = 23) |
|---|---|---|---|---|---|
| Formal | German | 7.95 (1.5) | 7.81 (1.2) | 6.83 (2.3) | 7.65 (1.4) |
|  | Italian | 2.20 (1.4) | 2.13 (1.6) | 3.34 (2.3) | 2.38 (1.9) |
| Informal | German | 5.12 (1.3) | 5.71 (1.9) | 5.26 (1.6) | 5.94 (2) |
|  | Italian | 5.54 (1.5) | 4.91 (2) | 5.76 (2.1) | 4.87 (2.4) |

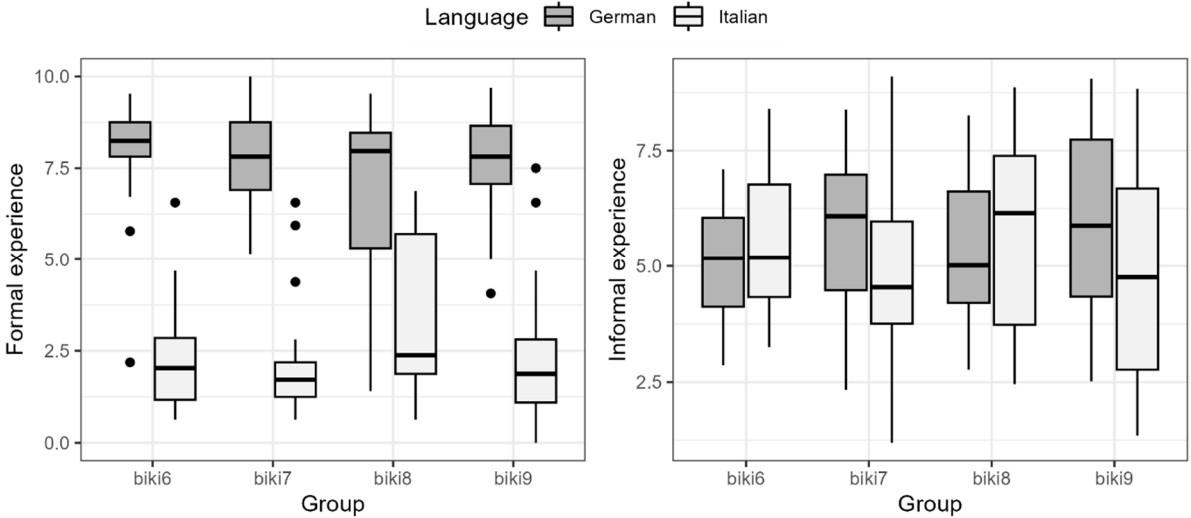

**Figure 2.** Boxplots depicting formal (**left**) and informal (**right**) language experiences (German and Italian).

For informal language experience, we found no effect of age for either of the three predictors (GE-inf: $p = 0.24$; IT-inf: $p = 0.61$); that is, informal language experience is stable across age groups (Supplementary Materials Section S4, Table S4.1).

## 5.2. Language Experience in Relation to Micro- and Macrostructure

Table 4 and Figure 3 summarize the results of the microstructure measures for each group in German and Italian, i.e., MATTR, speech rate, word-based MLU, and CI.

**Table 4.** Mean score (SD) of microstructure measures (German and Italian).

|  |  | Biki6 (*n* = 24) | Biki7 (*n* = 22) | Biki8 (*n* = 18) | Biki9 (*n* = 23) |
|---|---|---|---|---|---|
| MATTR | German | 0.95 (0.03) | 0.96 (0.02) | 0.96 (0.02) | 0.97 (0.01) |
|  | Italian | 0.92 (0.03) | 0.92 (0.04) | 0.92 (0.02) | 0.92 (0.03) |
| Speech rate | German | 1.87 (0.4) | 1.81 (0.5) | 1.98 (0.5) | 2.07 (0.4) |
|  | Italian | 1.68 (0.4) | 1.43 (0.7) | 1.98 (0.4) | 1.76 (0.5) |
| MLU | German | 7.37 (1) | 7.45 (1.3) | 8.04 (1.4) | 8.21 (1.3) |
|  | Italian | 7.49 (1.1) | 7.87 (1.8) | 8.71 (1.6) | 8.46 (1.5) |
| CI | German | 1.17 (0.2) | 1.16 (0.1) | 1.23 (0.2) | 1.29 (0.2) |
|  | Italian | 1.20 (0.1) | 1.27 (0.2) | 1.30 (0.1) | 1.30 (0.2) |

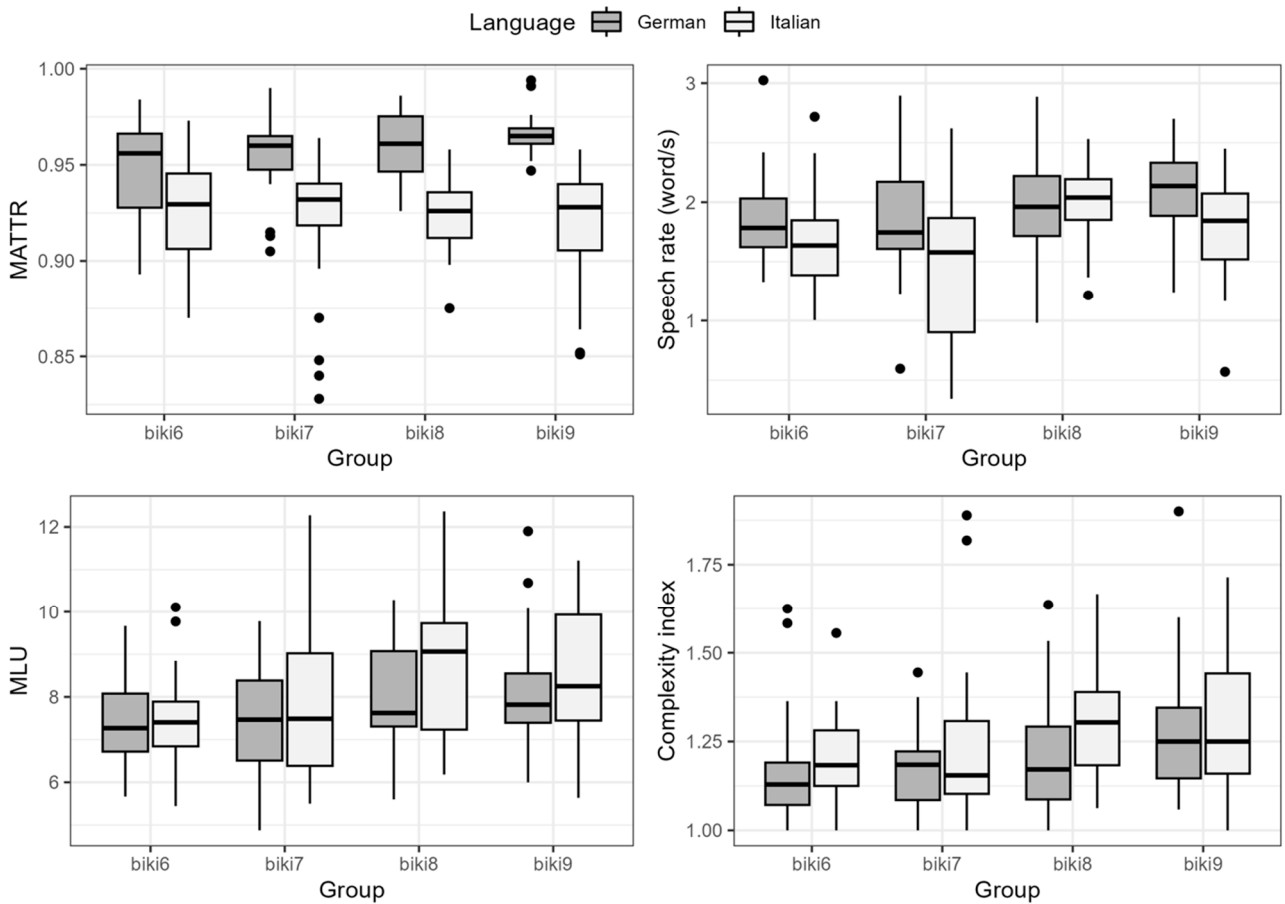

**Figure 3.** Boxplots depicting microstructure measures (German and Italian).

**Vocabulary diversity**[5]. MATTR in German was predicted by neither German or Italian informal experience (GE-inf: $p = 0.16$; IT-inf: $p = 0.16$) nor by formal experience (GE-for: $p = 0.62$; IT-for: $p = 0.73$) (Supplementary Materials Section S4, Table S4.4). For Italian MATTR, higher German informal experience predicted significantly lower MATTR scores (GE-inf: $p < 0.001$), while the effect of German formal experience was not significant (GE-for: $p = 0.75$). Higher Italian informal experience predicted significantly higher MATTR scores in Italian (IT-inf: $p = 0.001$), whereas Italian formal experience did not predict MATTR in Italian (IT-for: $p = 0.09$) (Supplementary Materials Section S4, Table S4.5).

**Speech rate**. Higher German informal experience predicted a significantly faster speech rate in German ($p = 0.002$), while German formal experience was not significant ($p = 0.08$). Conversely, higher Italian informal experience predicted significantly lower German speech rate ($p = 0.004$), whereas Italian formal experience did not predict speech rate in German ($p = 0.23$). Speech rate in Italian was significantly predicted by the amount of informal Italian experience and by informal German experience. In particular, higher Italian informal experience predicted a faster Italian speech rate ($p < 0.001$), while higher German informal experience predicted a slower Italian speech rate ($p < 0.001$). Formal language experience in Italian or German was not significant for Italian speech rate (GE-for: $p = 0.62$; IT-for: $p = 0.88$) (Supplementary Materials Section S4, Tables S4.6 and S4.7).

**Mean Length of Utterances**. MLU in German was not predicted by German (GE-inf: $p = 0.45$; GE-for: $p = 0.48$) or Italian informal and formal language experience (IT-inf: $p = 0.74$; IT-for: $p = 0.22$), but more informal experience in German led to a significantly lower MLU in Italian (GE-inf: $p = 0.047$). Furthermore, more informal experience in Italian also predicted a higher MLU in Italian, although not reaching significance (IT-inf: $p = 0.09$). However, formal language experience in German and formal language experience in Italian

did not have an effect on MLU in Italian (GE-for: *p* = 0.59; IT-for: *p* = 0.33) (Supplementary Materials Section S4, Tables S4.8 and S4.9).

**Syntactic complexity**. The CI in German was not significantly predicted by German formal (*p* = 0.31) and informal experience (*p* = 0.71), or by Italian formal (*p* = 0.29) and informal experience (*p* = 0.83). Similarly, Italian CI was not predicted by formal or informal experience in either language (GE-for: *p* = 0.86; GE-inf: *p* = 0.08; IT-for: *p* = 0.08; IT-inf: *p* = 0.31) (Supplementary Materials Section S4, Tables S4.10 and S4.11).

**Macrostructure**. Finally, Table 5 and Figure 4 summarize the mean and standard deviations (SDs) of the macrostructure measures for each group in German and Italian. The German and the Italian score B and score D were not predicted by the German or the Italian language experience in formal or informal contexts (score B-GE: GE-for: *p* = 0.99; GE-inf: *p* = 0.73; IT-for: *p* = 0.36; IT-inf: *p* = 0.79; score B-IT: GE-for: *p* = 0.99; GE-inf: *p* = 0.79; IT-for: *p* = 0.50; IT-inf: *p* = 0.60; score D-GE: GE-for: *p* = 0.79; GE-inf: *p* = 0.41; IT-for: *p* = 0.78; IT-inf: *p* = 0.12; score D-IT: GE-for: *p* = 0.19; GE-inf: *p* = 0.54; IT-for: *p* = 0.49; IT-inf: *p* = 0.88) (Supplementary Materials Section S4, Tables S4.12–S4.15).

**Table 5.** Mean score (SD) of macrostructure measures (German and Italian).

|  |  | **Biki6 (*n* = 24)** | **Biki7 (*n* = 22)** | **Biki8 (*n* = 18)** | **Biki9 (*n* = 23)** |
|---|---|---|---|---|---|
| Score B | German | 4.04 (1.4) | 3.50 (2.2) | 5 (2.5) | 5.22 (2.2) |
|  | Italian | 3.79 (1.9) | 3.59 (1.8) | 4.11 (1.9) | 4.52 (1.9) |
| Score D | German | 8.96 (1.2) | 9.18 (0.8) | 8.83 (1.2) | 9 (0.9) |
|  | Italian | 8.55 (1.2) | 8.95 (1) | 9.33 (0.8) | 8.96 (0.7) |

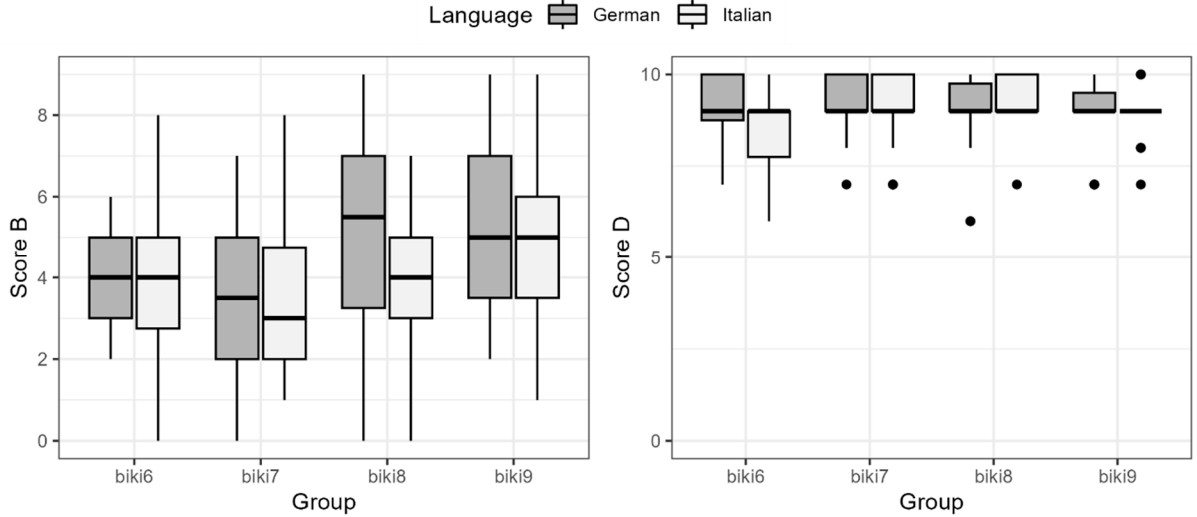

**Figure 4.** Boxplots depicting macrostructure measures (German and Italian).

The results, as relevant to RQ2, can be summarized as follows:

- Formal language experience showed no effects on any of the measures.
- There were no effects of language experience on syntactic complexity and any of the macrostructure measures.
- Vocabulary diversity was significantly predicted by informal language experience in both languages: more German experience predicted lower MATTR scores in Italian (*p* < 0.001), and more Italian experience predicted higher MATTR scores in Italian (*p* = 0.001).
- Speech rate was the most sensitive measure predicted by language experience: More German experience was related to slower speech in Italian (*p* < 0.001), and more Italian experience was related to higher speed in Italian (*p* < 0.001); more German experience was related to higher speech in German (*p* < 0.01), and more Italian experience was

related to lower speed in German ($p < 0.01$). This latter effect on German was not as strong as the aforementioned one of informal Italian experience on Italian speech rate.

- MLU was the only syntactic measure affected by informal language experience, with more informal German experience predicting lower MLUs in Italian ($p < 0.05$).

*5.3. Dimensions of Dominance (Experience and Proficiency) across Age Groups*

Finally, we tested whether dominance changed with age both in terms of narrative skills (micro- and macrostructure) and language experience. In the previous section, we focused on the two languages separately to test how the proficiency measures for each language are affected by language experience. Here, we look at the relative strength of the two languages: if a child has relatively more German than Italian experience, is her proficiency in German stronger than her Italian? Is this reflected in every linguistic domain? Does this differ across age groups? To do this, we calculated the differential scores and used them in the statistical models. Once we gained insights into how the two languages behave separately (Section 5.2), the differential scores provided an index of dominance that allowed us to look at how the two languages together are affected by language experience. Table 6 summarizes the differential scores of language experience and micro- and macrostructure measures for each group. Positive values indicate German dominance, while negative values indicate Italian dominance.

**Table 6.** Differential scores (SD) for language experience, micro- and macrostructure measures (German minus Italian).

|  | Biki6 (*n* = 24) | Biki7 (*n* = 22) | Biki8 (*n* = 18) | Biki9 (*n* = 23) |
|---|---|---|---|---|
| Formal | 5.75 (2.7) | 5.68 (2.4) | 3.49 (4.5) | 5.27 (3) |
| Informal | −0.42 (2.6) | 0.80 (3.8) | −0.50 (3.6) | 1.07 (4.4) |
| MATTR | 0.02 (0.03) | 0.04 (0.04) | 0.04 (0.03) | 0.05 (0.03) |
| Speech rate | 0.19 (0.6) | 0.37 (0.8) | 0 (0.4) | 0.31 (0.6) |
| MLU | −0.12 (1.1) | −0.42 (1.8) | −0.68 (1.5) | −0.25 (2.1) |
| Complexity Index | −0.04 (0.2) | −0.08 (0.3) | −0.07 (0.3) | −0.01 (0.3) |
| Score B | 0.25 (1.8) | −0.09 (2.8) | 0.89 (2.4) | 0.7 (2.5) |
| Score D | 0.46 (1.0) | 0.23 (1.1) | −0.5 (1.2) | 0.04 (1.1) |

**Vocabulary diversity**. MATTR differentials were significantly predicted by informal language differentials ($p < 0.001$) (Figure 5 right). A higher informal language differential score predicted higher MATTR differentials. In addition, age significantly predicted higher MATTR differentials ($p = 0.02$) (Figure 5 left). Formal language differentials were not significant ($p = 0.24$) (Supplementary Materials Section S4, Table S4.16).

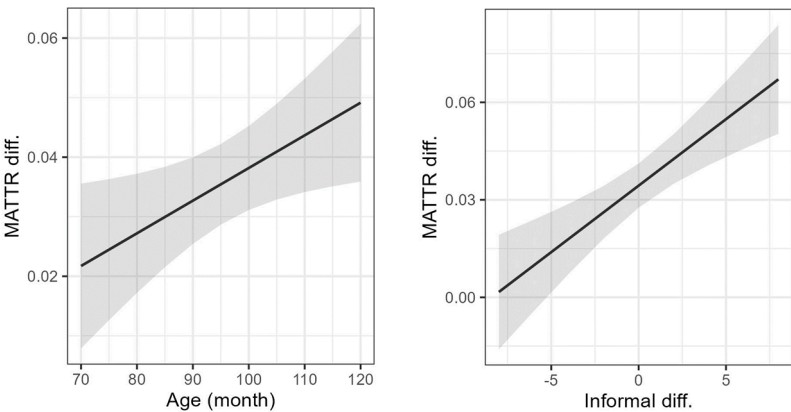

**Figure 5.** Effect of age (**left**) and informal differentials (**right**) on MATTR differentials.

**Speech rate**. Speech rate differentials were significantly predicted by formal language differentials ($p = 0.03$). Moreover, there was a significant interaction between formal

differentials and age ($p$ = 0.03). As shown in Figure 6 (top left), formal differentials had an effect on speech rate differentials, such that children with more formal experience in German had a relatively higher speech rate in German. However, this effect was only present in younger children (e.g., 70–80 months, corresponding to 6-year-olds), while being absent in the older children. In addition, a higher informal language differential score predicted higher speech rate differentials ($p$ < 0.001) (Figure 6 top center), pointing to dominance in the ML (Supplementary Materials Section S4, Table S4.17).

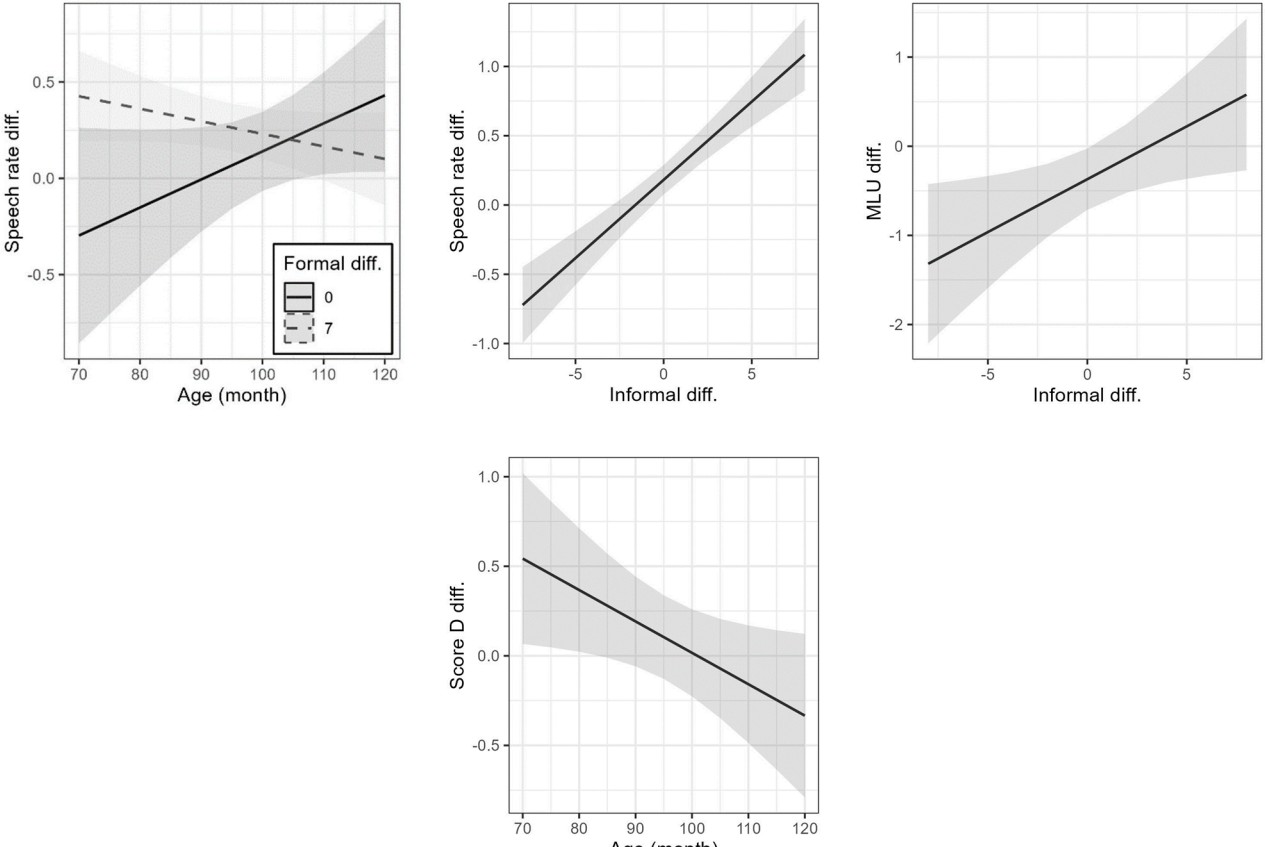

**Figure 6.** Interaction between age and formal differentials on speech rate differentials (**top left**), the main effect of informal differentials on speech rate differentials (**top center**) and MLU differentials (**top right**), and the main effect of age on score D differentials (**bottom**).

**Mean Length of Utterances**. The MLU differential score was not predicted by age ($p$ = 0.44) or formal language differentials ($p$ = 0.9), but it was positively predicted by informal differentials ($p$ = 0.02) (Figure 6 top right). This indicates that more experience with German (than Italian) leads to a comparatively higher MLU in this language (Supplementary Materials Section S4, Table S4.18).

**Syntactic complexity**. CI differentials were not predicted by age ($p$ = 0.86), formal ($p$ = 0.5), or informal language differentials ($p$ = 0.21) (Supplementary Materials Section S4, Table S4.19).

**Macrostructure**. Score B differentials were not predicted by age ($p$ = 0.41), formal ($p$ = 0.99), or informal language differentials ($p$ = 0.88). Score D differentials were similar in the two languages. They were significantly predicted by age, i.e., the older the children, the closer the differential score is to 0 ($p$ = 0.03), suggesting that older children are more balanced than younger children (see Figure 6 bottom). Formal ($p$ = 0.62) and informal language differentials ($p$ = 0.21) were not significant (Supplementary Materials Section S4, Tables S4.20 and S4.21).

In summary, we have treated the differential scores as a continuous variable to account for dominance, and we have investigated whether age and informal and formal language differentials predict micro- and macrostructure measure differentials. By and large, the results mirror those from the previous section (where age was not taken into account). Dominance in formal language experience only affected speech rate ($p < 0.05$), while dominance in informal language experience affected both speech rate ($p < 0.001$) and vocabulary (MATTR) ($p < 0.001$). Age was also relevant for these two language domains, i.e., speech rate ($p < 0.05$) and vocabulary ($p < 0.05$), with older children showing dominance in the ML. These tendencies were not reflected in macrostructure measures, where increasing age predicted more balanced score D differentials ($p < 0.05$)[6]. Overall, informal language experience and age were more powerful predictors than formal language experience, and vocabulary and speech rate were more sensitive to changes in these predictors than all other language domains.

## 6. Discussion

We investigated language dominance in six-to-nine-year-old Italian–German bilingual children. Our goal was to see how different degrees of formal and informal language experience (RQ1) correlate with measures of microstructure (proficiency) and macrostructure (RQ2). We calculated dominance differentials for all micro- and macrostructure measures and language experience to uncover potential differences across age groups (RQ3).

### 6.1. Formal and Informal Language Experience and Effects on Proficiency

First, we wanted to test whether language experience in the two contexts would change with age. With German being the ML, all groups have more language experience in formal contexts in German than Italian; therefore, we tested for differences within language but across age. We found that formal and informal language experience remained stable over time in both Italian and German in all age groups. However, the eight-year-olds had a slightly different profile than the other groups: they showed higher Italian experience in formal contexts because more children in this group attended schools with a bilingual curriculum or used Italian more often in formal contexts.

Second, we investigated how formal and informal contexts predict the chosen proficiency (microstructure) measures (RQ2). Previous research in bilingual development has repeatedly shown that language experience predicts acquisition, although the specific contributions of formal vs. informal exposure to the different language domains were still unclear. For microstructure, we considered lexical (MATTR), fluency (speech rate), morphosyntactic (MLU), and syntactic complexity (CI) measures, while for macrostructure, we considered story complexity (scoring B) and story comprehension (scoring D). Our main finding concerns language experience in informal contexts, which predicted all microstructure measures apart from syntactic complexity. In informal settings, Italian use had a positive effect on vocabulary and fluency scores in Italian, while German experience had a positive effect only on German fluency. A significant effect of language experience in informal contexts for vocabulary was also found by Mattheoudakis et al. (2016) and in adults, by Schmid and Karayayla (2020). Our findings confirm the importance of informal language experience for the ML and especially for the HL. Looking more closely at proficiency measures, speech rate seemed to benefit the most from informal language experience in both the ML and HL. This appears to indicate that informal interactions enhance fluency more effectively than a formal setting (e.g., school), arguably due to the variety of interlocutors, the amount of time spent with them, and the relaxed atmosphere where one can "try out" language without feeling any pressure.

Across languages, we found that the German informal language experience resulted in lower scores for Italian, vocabulary, speech rate, and morphosyntax while, conversely, the Italian informal language experience caused lower scores in German with respect to speech rate. As to cross-language effects, we are wary of attributing this bidirectional influence on a negative effect of one language on the other, because the scoring in the two

languages was not always independent from each other; in particular, for the informal measures, the score calculation had a high degree of complementarity since some parts of the German score were obtained by subtraction from the Italian score. Instead, these results can be interpreted in a positive way, such that more informal language experience favors linguistic development, especially in the HL. Finally, we observed that syntactic complexity, including in the HL, was the least sensitive measure of language experience. This is in line with earlier comparisons of syntax and vocabulary in much younger children (e.g., Paradis and Genesee 1996), and it might suggest that syntactic complexity develops and stabilizes earlier than school entry, thus being less vulnerable. It is possible that more fine-grained measures based on individual syntactic phenomena would be more sensitive than the number of subordinate sentences, as analyzed here.

Based on previous research (e.g., Golberg et al. 2008; Paradis 2011; Jia and Paradis 2015; Mattheoudakis et al. 2016; Bayram et al. 2019), we expected to find effects of formal language experience, especially for the development of vocabulary and syntax. However, formal language experience showed no effects on any of our proficiency measures, neither in German nor in Italian. There are three possible explanations for the absence of an effect of formal experience: (i) the role of formal language experience has been overestimated in previous research and does not contribute as much as informal experience, (ii) the language experience measure we used was not sensitive enough to bring out effects of formal language experience, and effects of formal experience might have emerged if we had taken cumulative experience into account (see Mattheoudakis et al. 2016), and (iii) there is very little individual variation in formal language experience in the two languages of our groups because, with the exception of the Biki8 group, most children received formal education only in German. If formal language experience is high in German and low in Italian for most children, the variability observed in the measures of microstructure cannot be explained by the children's formal language experience. Based on our data, we cannot tease apart these three possibilities.

Finally, for macrostructure, no effects of language informal or formal experience were found, which is in line with Bohnacker et al. (2021) and Lindgren and Bohnacker (2022). This is expected because macrostructure does not involve language-specific skills but taps into more general cognitive skills, so children tend to perform similarly irrespective of language.

In summary, formal and informal language experience remained stable across age groups during primary school. Informal language experience was the only type of language experience that showed effects within and across languages, suggesting that exposure in informal contexts to the ML and, in particular, to the HL, is crucial for the development of language proficiency during primary school.

*6.2. How Does Dominance Change?*

The findings of RQ1 and RQ2 were prerequisites for answering RQ3. RQ1 and RQ2 informed us about formal and informal language experiences, and their impact on specific linguistic domains. RQ3 aimed to investigate whether the difference between proficiency in German and Italian changes with age and/or differences in formal and informal language experience. To address this question, we calculated differential scores for each micro- and macrostructure measure by subtracting the Italian from the German score.

Also, in this case, the most informative measure was language experience in informal contexts. Changes in informal language experience explained many changes in microstructure dominance in our data. Moreover, the directionality of the effect was the same for vocabulary, fluency, and morphosyntax: Whenever children had more informal experience in the ML, this resulted in higher proficiency in the ML. This suggests that if language experience in informal contexts is predominantly in the ML, this can drive a shift towards the ML in all language domains.

The effects of formal language differentials, by contrast, were limited to fluency, and only appeared in younger children; there were no effects on vocabulary, where we had

expected the effects to be the strongest (Mattheoudakis et al. 2016). As discussed earlier, the limited effects of formal language experience in our study could have methodological reasons; alternatively, previous studies might have overestimated the role of formal as opposed to informal experience. Future research should look more closely at formal and informal exposure cumulatively and, ideally, include children having a wide range of formal language experience in their two languages. Also unexpectedly, syntactic complexity was not affected by formal experience and/or age. One possible explanation for this is that the retelling mode of the task is beneficial to the production of complex sentences. However, an effect of language experience dominance on syntactic complexity was found in other studies using a narrative task in the retelling mode (Andreou et al. 2015); therefore, we would exclude a task effect in this case. Moreover, it has been shown that story retelling can elicit longer and more accurate stories (Hayward et al. 2007). Interestingly, the syntactic complexity measure (in line with the MLU) generally yielded higher values in the HL Italian than ML German, irrespective of age. We have speculated above that this might be a result of syntactic complexity developing earlier and thus being more robust to changes in linguistic experience. Alternatively, it could reflect stylistic preferences in the two languages.

The results on macrostructure did not show any effects or interactions with language experience. This can be explained by the fact that macrostructure is not language-specific. Instead, there was an effect of age for story comprehension differentials (scoring D), suggesting that the older children have become more balanced in comprehension, thus being equally good at answering questions in both languages. Overall, the children answered the comprehension questions successfully in both languages, in line with previous studies (Bohnacker and Lindgren 2021; Bohnacker et al. 2021).

Can we see dominance changing across age? Our findings do not show strong effects of age alone, and certainly no shift, as the latter would have implied a reversal from Italian as the dominant language to German as the dominant language. Instead, the younger children were fairly balanced, and while Italian remained stable, German proficiency grew for vocabulary and speech rate. Moreover, we found a joint effect of age and language experience, suggesting that language experience is a driving force for a widening imbalance. Importantly, this was not present in all measures we used, but only for vocabulary and fluency.

## 7. Conclusions

We investigated language dominance in Italian-German bilingual children using three sets of data: questionnaire-based language experience data and narrative microstructure and macrostructure data. Children had similar amounts of formal and informal experience in German (ML) and Italian (HL) across ages. We explored the relationship between language experience and proficiency (microstructure) measures and found that, although previous research has stressed the role of formal language experience on linguistic skills (particularly vocabulary), we found no strong effects of formal experience. Instead, it was informal language experience that played a significant role in most linguistic domains. Syntactic complexity appears to be an exception, arguably because much of it develops and stabilizes before primary school. Our findings have societal and educational implications: Despite the decrease in formal exposure, which might inevitably happen and is harder to counter, the development of the heritage language can be supported by a continuous and varied language experience in informal settings. Moreover, we investigated whether dominance changed in relation to language experience and age. We did not observe a shift in dominance, possibly because the shift had already occurred when children were younger. Overall, our study provides evidence to encourage the use of the HL in informal contexts to promote its linguistic development and maintenance.

**Supplementary Materials:** The following supporting information can be downloaded at: https://www.mdpi.com/article/10.3390/languages9020063/s1, Section S1: Questionnaire on the child's linguistic history, Section S2: Questionnaire scoring, Section S3: Additional (descriptive) results of individual questions, Section S4: Statistical models (Lüdecke 2022; R Core Team 2021).

**Author Contributions:** Conceptualization, M.P.; data curation, M.P., M.F.F., N.F. and M.G.; formal analysis, M.F.F. and M.G.; funding acquisition, T.M. and T.K.; methodology, M.P., M.F.F., N.F. and M.G.; supervision, T.M. and T.K.; writing—original draft, M.P.; writing—review & editing, M.P., M.F.F., N.F., M.G., T.M. and T.K. All authors have read and agreed to the published version of the manuscript.

**Funding:** This research was funded by the Deutsche Forschungsgemeinschaft (DFG, German Research Foundation) as part of the project KU 2439/5-1 'Non-Canonical Questions in Early and Late Bilingual Language Acquisition' within the research unit FOR2111 'Questions at the Interfaces'.

**Institutional Review Board Statement:** The study was conducted in accordance with the Declaration of Helsinki, and approved by the Institutional Review Board (Ethics Committee) of the University of Konstanz (IRB Statement 14/2020, 4 August 2020).

**Informed Consent Statement:** Informed consent was obtained from the parents of all subjects involved in the study.

**Data Availability Statement:** The data presented in this study can be found here: https://osf.io/f6nxy/.

**Acknowledgments:** We thank all the parents and children who took part in the study. We thank Sebastiano Arona, Alexandra Besler, Carolin Hennerich, Filippo Graziano, Chiara Ochsenreiter and Laura Steck for help with data collection, transcription and annotation. We also thank the reviewers for their comments and suggestions.

**Conflicts of Interest:** The authors declare no conflict of interest.

## Notes

[1] Within Biki6, 15 children still attended kindergarten. Nevertheless, we label all the children in this study as "primary school children" since they were all attending or about to start primary school. In fact, we did not include any children younger than 6 years old.

[2] The complete questionnaire is provided in Supplementary Materials Section S1. The questionnaire includes two separate sections that tackle child education before/after vs. during the pandemic since children were tested at different stages of the pandemic. For the analysis, we calculated a unique average score for these sections to have a more reliable view of the child's experience with and without lockdowns.

[3] According to score D, the children in our study struggled to give the correct answer to the tenth question, which asked whether the child would become friends with the cat/dog after it stole the child's fish/sausages. According to the manual, the answer is *no*, but the children seemed to find it hard to think that the child could not become friends with the animal. To ensure comparability across studies, we followed the manual and gave 0 points for *yes*-answers, despite the plausibility of the children's motivations.

[4] Thirteen children in the Biki6 group, ten in Biki7, six in Biki8, and seven in Biki9 attended monolingual schools. The number of children attending monolingual schools and Italian (extracurricular) classes was nine in Biki6, nine in Biki7, five in Biki8, and eleven in Biki9. The number of children attending bilingual schools was two in Biki6, three in Biki7, seven in Biki8, and five in Biki9.

[5] For each measure in this section, we calculated separate models for each language.

[6] Regarding the effect on score D, our reservations on the interpretation are discussed above.

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
