# Peer review of "On the Role of Informal vs. Formal Context of Language Experience in Italian–German Primary School Children"

_languages, doi:10.3390/languages9020063_

Round 1

Reviewer 1 Report

Comments and Suggestions for Authors

Reviewer 2 Report

Comments and Suggestions for Authors

The goal of the paper is to understand the pathway through which language dominance develops/shifts in school-age multilingual children, and what difference it makes if formal or informal language experience is considered. The goals were clearly presented, relevant data and methods introduced and tested, and conclusions clearly follow. 

I particularly like the closing message that this study provides additional evidence supporting claims that children's majority language knowledge will not be hampered by heritage language use in informal contexts. Yet another important "public service announcement," if it can be appropriately disseminated.

One caveat is that my own research area is NOT in language development, so I can only accept the authors' word that they have presented relevant and related research but that their own work, as specified in good detail in the paper, is original. I have no reason to doubt it. 

I have two suggestions: 

1) Explain more clearly what is gained by looking at the "differential scores" rather than only the language-specific scores.  Can you illustrate to what extent , for example, "formal experience in Italian" is not a mirror image of "formal experience in German."  Or, if one method is more revealing than the other, why not present only that one, or at least say why it's better?  (You hint at this very late, in the discussion on p. 16, saying these are "not always independent measures."

2) in lines 609-612, it is suggested that the syntactic complexity measure yields higher values in Italian than German and that it could "reflect stylistic preferences." Could it also just be a measure that gives higher values in Italian than German given the languages' structures.  Perhaps check "comparative complexity" literature like:

Markus Sadeniemi , Kimmo Kettunen , Tiina Lindh-Knuutila & Timo Honkela (2008) Complexity of European Union Languages: A comparative approach, Journal of Quantitative Linguistics, 15:2, 185-211, DOI: 10.1080/09296170801961843

Koplenig, Alexander, Peter Meyer, Sascha Wolfer & Carolin Mueller-Spitzer.

2017. The statistical trade-off between word order and word structure—

large-scale evidence for the principle of least effort. PloS one 12(3). e0173614.

Very small things:

line 244 - groups referred to as "biki" . Does this word have a meaning that would be useful to know?

line 330 and following - I found the use of the term "differential" confusing, when just "difference", as in subtraction, rather than change in velocity (or something else calculus-like to which I associate "differentials") could be used. 

In Figure 6a, add "age" to the legend to clarify what the 2 lines represent. 
